# Detection of Superior Rice Genotypes and Yield Stability under Different Nitrogen Levels Using AMMI Model and Stability Statistics

**DOI:** 10.3390/plants11202775

**Published:** 2022-10-19

**Authors:** Mohamed Abdelrahman, Khadiga Alharbi, Medhat E. El-Denary, Taher Abd El-Megeed, El-Sayed Naeem, Samah Monir, Najla A. Al-Shaye, Megahed H. Ammar, Kotb Attia, Said A. Dora, Abdel-Salam E. Draz

**Affiliations:** 1Rice Research and Training Center, Field Crops Research Institute, Agricultural Research Center, Kafrelsheikh 33717, Egypt; 2Department of Biology, College of Science, Princess Nourah Bint Abdulrahman University, P.O. Box 84428, Riyadh 11671, Saudi Arabia; 3Genetics Department, Faculty of Agriculture, Tanta University, Tanta 31527, Egypt; 4Center of Excellence in Biotechnology Research, King Saud University, P.O. Box 2455, Riyadh 11451, Saudi Arabia; 5Genetics Department, Faculty of Agriculture, Kafrelsheikh University, Kafrelsheikh 33716, Egypt

**Keywords:** rice, sustainable, low input, nitrogen fertilizer, AMMI, grain yield, stability, multi-location, multi-environment

## Abstract

Sustainable agriculture is a prerequisite for food and environmental security. Chemical fertilization, especially nitrogenous fertilization, is considered the most consumed for field crops. In rice crops, plants consume much less than half of the applied N-fertilizer. In the current investigation, multiple N environments were generated by applying different N doses of urea fertilizer to a permanent transplanted field for two successive summer growing seasons at the rice research and training center, Kafrelsheikh, Egypt. A set of 55 genotypes consisting of 25 Jabonica, 4 Tropical Japonica, 20 Indica, and 6 Indica/Japonica were transplanted under no N (0N), Low N (LN), medium N (MN), and High N (HN) (i.e., 0, 48, 96, and 165 Kg N ha^−1^, respectively). Highly significant differences were detected among the tested genotypes. AMMI analysis of variance revealed the existence of the genotype via nitrogen interaction (GNI) on yield performance. The GNI principal components (IPCA); IPCA1 and IPCA2 scores were significant and contributed values of 71.1 and 21.7%, respectively. The highest-ranked genotypes were MTU1010, IR22, SK2046, SK2058, IR66, and Yabani LuLu based on their grain yield means (30.7, 29.9, 29.5, 29.3, 28.8, and 28.5 g plant^−1^). These genotypes were grouped into the same subcluster (SCL) according to the stability analysis ranking matrix. Based on AMMI analysis and biplots, MTU1010 and Yabani LuLu showed yield stability across environments. Meanwhile, the which-won-where biplot showed that IR22 was superior under unfavorable N-levels and MTU1010 was stable across the different environments. These findings are considered to be of great importance to breeders for initiating low-nitrogen-input breeding programs for sustainable agriculture.

## 1. Introduction

Rice (*Oryza sativa* L.) is a staple food in many countries and a major component of diets in many others. In many parts of the world, a lack of rice supplies could lead to starvation. Globally, 755 million tons of paddy rice are produced from 162 million ha of land [1]. To meet the needs of the world’s rapidly growing population, increasing grain yield per unit area of rice is critical to combating poverty. This goal could be met by cultivating rice cultivars with high yield potential and using appropriate management [2].

Nitrogen (N) is the most important nutrient that is needed by plants for growth and completing their life cycle. However, the rice crop consumes much less than half of the applied N fertilizer. The recovery efficiency of nitrogen fertilizer in rice amounted to 30–39%, and the nitrogen use efficiency decreases with the increase in the applied nitrogen [3]. N application losses contribute to soil deterioration, groundwater pollution, and emissions of ammonia and greenhouse gases [4,5]. Accordingly, for rice production sustainability that is less harmful to the environment, it is necessary to identify the superior genotypes under low application of N-fertilizer. Hence, rice breeders are always evaluating a large set of genotypes under different environments to identify the high-yielding genotypes that have adapted to different environments [6,7]. 

Grain yield (GY) is the economic determinant of the best-performing genotypes [8]. GY is a quantitative trait determined by the additive main effect of environment (E) and genotype (G) in addition to the nonadditive effect of the G X E interaction (GEI) [9]. Breeders focus on the GEI effect to identify the yield stability of genotypes across different conditions and environments, which cannot be revealed by the separate effects of genotype or environment [10,11,12,13,14]. The GY heritability is exposed to variability across different environments [15,16], which hinders the accuracy of superior varietal selection processes [17]. Therefore, widely adapted genotypes with the ability to produce stable high yields across diversified environments constitute a major goal for rice breeders. Hence, it is critical to evaluate the adaption of several genotypes across different N inputs to identify superior genotypes with stable GY under different N environments to detect GNI. Furthermore, the analysis of yield stability under different soil conditions is found to be associated with imbalanced yield stability [18]. Several statistical analysis methods have been reported and developed to manipulate GEI including parametric and non-parametric stability statistical methods. Parametric methods include univariate and multivariate methods. Wricke’s ecovalence (Wi2) [19], Shukla’s stability variance (s2) [20], the coefficient of variance (CV) [21], Environmental variance (S2) [22], the Mean-variance component (q) [23], the GE variance component (q’) [24], the Regression coefficient (bi) [25], and many others constitute univariate methods. Multivariate methods include the additive main effects and multiplicative interaction (AMMI) model [26] and the GGE biplot method [27]. AMMI analysis considers both ANOVA and principal component analysis (PCA). The results of AMMI provide genotypes’ yield stability under different environments and facilitate the precise selection of the best-performing genotypes for the environment under study [28,29,30]. Multivariate methods estimate GEI by following approaches such as the ‘which-won-where’ pattern, identifying mega environments and superior genotypes across different environments under study, and ranking environments [31]. Non-parametric methods include Nassa and Huhn’s and Huhn’s statistics (S) [32], Kang’s rank-sum (KR) [33], TOP-Fox (TOP) [34], Thennarasu’s non-parametric statistics (NP) [35], and the Genotype stability index (GSI) [36]. Recently, a new R package has been reported to manipulate the multi-environment trials (MET), called metan [37]. The package is a work-flow based approach that has a collection of functions to compute the most used parametric and nonparametric stability statistics. Unlike other R software packages that could possibly be used for analyzing MET data, metan is specifically coded for a complete analysis of MET trials (checking, manipulating, analyzing, and visualizing the data).

In recent years, there has been substantial interest in detecting both the yield and yield stability under different fertilizer conditions in different crops [38]. In our recent investigation, we estimated the yield stability of newly developed rice lines across different water treatments [15]. The present study aims to identify superior rice genotypes under low nitrogen input with stable yield performance over different N levels. The study is expected to enrich our knowledge regarding low-nitrogen-input genotypes with high yielding ability.

## 2. Results

### 2.1. Genotypic Variability of GY under Different N Environments

The mean performance of the evaluated genotypes under different N environments is presented in Appendix A. The data indicate that the genotypes exhibited different behavior as a response to the different N treatments, which was confirmed by the analysis of variance as indicated from the genotypic mean square value under different N levels (Appendix A). The mean performance of the tested genotypes across the different N environments showed that MTU1010 has the highest GY mean value across N environments, with 30.7 g plant^−1^. Moreover, environments’ mean performances showed an increase in the overall GY mean performance of the genotypes with each incremental dose of N, which ranged between 19.14 and 28.84 g plant^−1^ for the 0N and HN levels under study. 

### 2.2. Combined AMMI Analysis of Variance of the GY and the Decomposition of GEI Effect

The AMMI model analysis for the genotypes’ GY revealed significant differences for both the main (G and E) and GEI effects (Appendix A). These findings show that there is considerable variability among the genotypes, environments, and their interactions. The G factor accounted for 49.97% of the total variance in the AMMI analysis, followed by E (33.69%) and GEI (7.40%). The decomposition of the GEI effect by AMMI analysis yielded three significant IPCAs. These three IPCAs explained 100 percent of the total GEI effects explained, which were all significant accounting for 71.1%, 21.7%, and 7.2% for IPCA1, IPCA 2, and IPCA3, respectively.

### 2.3. Environmental Effect on the Performance of the Genotypes

Based on the output of the AMMI analysis, Table 1 presents the mean genotypic GY performance in each N-environment, EPCA1, and the top five ranked genotypes for each N environment. The environments had different records of the overall genotypes’ GY mean. According to the environmental index the environments were classified into unfavorable and favorable for the grown genotypes under these environments. Environments with negative index estimates were unfavorable while MN and HN were favorable. 0N was the most unfavorable while HN was the most-favorable condition for the genotypes under study. Furthermore, according to the estimated EPC1, the results showed that N-environments contributed differently to the genotype stability for the GY. Among the different N- levels, 0N has the lowest PC1 estimate while HN has the highest PC1 record. These records indicate that the HN was the main contributor to the genotypic GY stability and 0N has the lowest contribution to the GEI.

### 2.4. Graphical Representation of Genotypes and N-Environments in the AMMI Biplots

The biplots for AMMI1-Means vs. PC1 and AMMI2-PC1 vs. PC2 were generated to identify the mega environment and stable genotypes across N treatments (Figure 1A, Appendix A). The AMMI1 biplot was generated to plot genotypes’ and environments’ means against their IPCA1. Subsequently, the environments 0N and LN were identified as low-yielding ones (Figure 1A AMMI1 Biplot). The AMMI2 biplot reveals the environmental and genotypical scores corresponding to both the GY IPCA1 and IPCA2 (Figure 1B AMMI2 Biplot). Based on this plot, the environments and/or the genotypes that are located in close proximity to the origin are less influenced by the effect of GEI. On the contrary, those genotypes and/or environments that are distanced from the origin either on the negative or positive side are more influenced by the GEI effect. Among the different N environments, LN was the nearest to the GEI origin compared with all other environments. Meanwhile, on the genotypes side, C22, WAB 450-1-B-P-91-HB, Black Rice, Arabi, Yun Lu No. 48, GZ 6522-15-1-1-3, Giza178, SKC 23808-28-5-2-1-1, Taikeng Yu 1420, IR28, Sakha104, and SK2058 generated a polygon where they were the most interactive genotypes with the changes in N levels.

### 2.5. Which-Won-Where GGE Biplot Analysis

The sum of the first and second PCA axes explained 96.36% of the total GEI variation, as can be seen in the GGE biplot presented in Figure 2A. The genotypes that were farthest away from the biplot origin include Black Rice, Yun Lu No. 48, Arabi, IR22, MTU1010, Giza178, Reiho, SKC 23808-28-5-2-1-1, and Giza177. These genotypes, called vertex genotypes, have the longest vectors with respect to their direction. Vertex genotypes are the most responsive genotypes to the environment in their direction. The vertex genotypes with no environmental indicators nearby are the poorly performing ones. Accordingly, Black Rice had the poorest performance in all environments under study. The which-won-where GGE biplot of GY divided the four N treatments into two sectors. The low treatments, 0N and LN, were located together in one sector where the IR22 genotype was the best-performing genotype, while MN and HN were located together in the other sector. MTU1010 showed a stable performance under different N treatments, placing it in the top five ranked genotypes across the tested N environments.

### 2.6. Superior Genotypes Selection Based on GY Means and Stability Parameters

The ranking based on the various stability statistics is presented in Appendix A. Based on GY, which considers the main selection criteria for genotype selection, the genotypes MTU1010, IR22, SK2046, SK2058, and IR66 were the highest-ranked genotypes in this regard. From another stability statistics analysis, the weighted average of absolute scores (WAASB) was used to better identify the best genotypes based on the mean GY and stability. The biplot shows the distribution of the tested rice genotypes and environments based on the genotypes’ GY mean and WAASB values, as shown in Figure 2B.

The first quadrant, I, contains the genotypes Black Rice, WAB 450-1-B-P-91-HB, Yun Lu No. 48, Sabieny, Pusa Basmati 1, Giza177, SKC 23808-28-5-2-1-1, GZ 6522-15-1-1-3, GZ5830-59-10-2, Taikeng Yu 1420, and IR 67075-2B-5-2. These genotypes showed lower grain yield compared to the mean grain yield. In addition, this quadrant has low GY environments 0N and LN. Accordingly, the genotypes and environments located in this quadrant have the largest response to GEI. The second quadrant contains the environments with GY above average, MN and HN environments, and the genotypes that have a GY above average with a high GEI response, namely, Arabi, Reiho, Giza178, SK2034, WAB 880 SG 73, Giza14, Nahda, and IR22. The genotypes GZ7718-13-3-1-3, SKC23822-304-3-1-1-1, GZ6214-4-1-1-1, GZ7718-13-3-2-2, IR68353-35-3-3-2-2-1-2, IR68373-R-R-B-22-2-2, Yen Geng 135, Agami M.1, IR 73689-31-1, Sakha103, Giza182, IR70, and GZ6903-3-4-2-1 showed low yield but stable performance across the different environments. These genotypes were located in the third quadrant and there were no environments in this one. The fourth quadrant contains the high-yielding stable genotypes. Those genotypes are MTU1010, SK2046, SK2058, GZ7922-B-44-1, Egyptian Yasmin, Yabani LuLu, IET 1444C22, SK2035, BG 304, IR 74, Milyang63, GZ6910-28-1-3-1, IR74, IR64, GZ6903-3-4-2-1, GZ6910-28-1-3-1, E 7034, and Nabatat Asmar.

### 2.7. Cluster Analysis and Dendrogram Based on the Stability Statistics Values

A hierarchical cluster analysis was conducted based on the squared Euclidean distance via Ward’s method using values of the stability statistics for each genotype. The grouping pattern resulting from this analysis revealed the distribution of the tested rice genotypes into two main clusters (CL) (Figure 3). CL-1 was further divided into two subclusters (SCL). SCL1 contains the genotypes WAB 450-1-B-P-91-HB, Black Rice, Arabi, and Yun Lu No. 48, while SCL2 has the genotypes Sabieny, IR 67075-2B-5-2, GZ7922-B-44-1, Sakha103, Giza175, GZ6214-4-1-1-1, IR7421-35-1-1-2, GZ7718-13-3-2-2, IR74, Agami M.1, and SKC23822-304-3-1-1-1. At the same time, CL2 was divided into two main SCLs, SCL3 and SCL4. SCL3 was divided into two groups, SCL3-1 and SCL3-2. SCL3-1 contains the top six ranking genotypes, MTU1010, IR22, SK2046, SK2058, IR66, and Yabani LuLu, with the highest GY mean across environments. These genotypes are indica, indicating that indica-type genotypes have more adaptability to withstand different N doses and provide high yield ability. Meanwhile, the genotypes Taikeng Yu 1420, Giza178,48, BG 304, IET1444, IR70, SK2034, C22, SK2035, Giza14, and Egyptian Yasmin were clustered in SCL3-2. Similarly, SCL4 has two main clusters, SCL4-1 and SCL4-2. The genotypes Pusa Basmati 1, Milyang63, Nabatat Asmar, GZ6903-3-4-2-1, Giza159, and Nahda clustered in one group together with the other group of IR64, Sakha101, GZ6910-28-1-3-1, E7034, Sakha104, WAB880 SG 73, Yen Geng 135, IR68373-R-R-B-22-2-2, IR73689-31-1, Giza182, and IR28. Both groups were clustered from SCL4-1. Meanwhile, the genotypes GZ5830-59-10-2, IR 68353-35-3-3-2-2-1-2, Giza177, GZ 6522-15-1-1-3, and SKC 23808-28-5-2-1-1 were grouped in SCL4-2.

## 3. Discussion

GY is the main criterion for a breeder based on which new genotypes are selected. Basically, it is the final product of the genotype performance, which considers the output of GEI. In recent years, great attention has been paid to sustainable cultivation. It mainly focuses on reaching the maximum yield potential of a genotype with a lower input of water, chemical fertilizers, and pesticides [4,15]. 

In the current investigation, considerable variability was detected among the tested genotypes based on their GY performance under different N levels. The genotypes have different response patterns for each incremental dose of N fertilizer (Appendix A and Appendix A). The AMMI-ANOVA analysis for the combined two-season data revealed highly significant differences between genotypes, N-environments, and GEI. However, the proportion for the genotypic variability was the highest. This finding explains the existence of selection capacity to find better-performing genotypes under different N levels since all other elements were consistent except for the application of N- chemical fertilizer. This suggestion is further confirmed by our findings of an interaction between the genotypes and the different N levels (GEI) regarding the amount of variability. However, several investigations confirmed the findings of significant variability due to G, E, and GEI, but the environments had the highest source of variation (i.e., [39,40]). These trials were conducted at different locations, which means the environmental conditions are entirely different from each other. In our investigation, the environments 0N and HN were the lowest and highest GY mean records, indicating the possibility of further enhancing the varietal productivity under low-N-input environments. These findings confirm the need for stability analysis for the tested genotypes under multi-N-levels. 

Among the widely used stability analysis methods, the AMMI model combines both ANOVA and GEI to measure the magnitude of genotype and environment variability [26,41]. In our investigation, a significant interaction between N fertilizer and the tested genotypes in GY, as a high proportion of the first two IPCAs, was found. Variability in response to GEI identifies that genotypes with different ranks of GY potential correspond to specific environments and their stability across the environments [42]. For the tested environments, high N levels were favored by the genotypes. However, genotypes that efficiently developed high yields while growing under poor N levels have the ability to tolerate N deficiency in the soil. Genotypes IR22, Giza14, C22, MTU1010, and SK2058 were the highest yielding under the 0N environment. Interestingly, those genotypes are the indica type, except Giza14, which is the japonica type. Furthermore, these genotypes were comparable to the newly developed cultivars under this level. The AMMI biplots were used to measure the distribution of G, E, and GEI between them [43], as indicated in the AMMI1 and AMMI 2 biplots shown in Figure 1A,B. The AMMI1 biplot explained the distribution of the tested genotypes based on the GY mean and IPCA1 values, which are useful to identify the genotypes MTU1010, IR66, and Yabani LuLu with the lowest IPCA1 scores and a high GY mean. Furthermore, the tested environments exhibited wide dispersion relative to the GY performance of the genotypes under study. The AMMI2 biplot further described the multiplicative effects of GEI through the first two IPCAs (Figure 1B). Accordingly, the genotypes G25, MTU1010, BG 304, and Yabani LuLu were the best genotypes having the lowest IPCAs values and located in close proximity to the origin. Using AMMI biplots, several investigations were able to identify their corresponding genotypes such as in barley [44], rice [40], Sorghum [45], cassava [46], and maize [47]. MTU1010 and Yabani LuLu are among the two groups most productive and stable. Intriguingly, IR22 showed specific adaptability to the low N environments (0N, LN, and MN), which was further confirmed by the which-won-where GGE biplot, as shown in Figure 2A. Moreover, MTU1010 was located at the boundary between the two sectors where the four environments were scattered, indicating that this genotype has stable performance across the different N-environments. Both IR22 and MTU1010 are indica genotypes, while the study has other types such as japonica and indica/japonica types (Table 2). IR22 is the cultivar derived from the miracle rice, IR8. MTU1010 is an elite, mega Indian cultivar grown extensively in India [48]. These findings may indicate that indica type has better performance than other types under study environments, which may be due to high grain nitrogen use efficiency [49]. This point requires further analysis and investigation.

We further estimated the stability measure of the weighted average of the absolute score (WAASB), which was taken into consideration as the sum of absolute values of the IPCAs [50]. To identify the highly adaptive and best-performing genotypes, we generated a biplot to simultaneously consider the GY and stability measure of WAASB. Among the tested environments, the two MN and HN environments were productive with average GY means higher than the grand mean. The genotypes MTU1010, IR66, Yabani LuLu, and GZ7922-B-44-1, with high GY and the lowest WAASB values, were identified as the most stable genotypes across the different N environments under study. The WAASB index was proven to be an efficient index in the selection of stable, high GY barley genotypes across different environments [51,52].

To identify groups of the genotypes based on the different stability statistical analysis ranking matrices, CLA was conducted. The most notable findings were that the low-yielding genotypes WAB 450-1-B-P-91-HB, Black Rice, and Yun Lu No. 48 were grouped in a single SCL while the high-yielding ones, IR22, SK2046, SK2058, MTU1010, Yabani LuLu, and IR66, were grouped in another SCL. The former group was low yielding under favorable and unfavorable N environments, while the latter group of genotypes was high yielding under all environments.

Considering our results, MTU1010 and Yabani LuLu showed high performance and GY stability across the different N-levels, while IR22 showed superior performance under the unfavorable N-levels. Our results need more investigations regarding the performance of these genotypes with respect to nitrogen use efficiency-related parameters.

## 4. Materials and Methods

### 4.1. Plant Materials

To maintain diversity, a set of 55 rice genotypes consisting of 25 Jabonica, 4 Tropical Japonica, 20 Indica, and 6 Indica/Japonica were used in the current experiment (Table 2). Among these genotypes were 15 old Egyptian rice cultivars. These genotypes were collected from rice GeneBank unit of the rice research and training center (RRTC), Kafrelsheikh, Egypt.

### 4.2. Experimental Location and Soil Properties

The 55 rice genotypes were evaluated for two successive seasons at the Sakha Agricultural Research Station experimental farm (31.09° N and 30.9° E), located in Kafrelsheikh, Egypt. During the two seasons, the climate was typified by high temperatures and no rainfall. The soil analysis was conducted according to Piper [53] and Black et al., [54], where the soil texture of both seasons was clayey with a total N of 759 and 770 mg kg^−1^ for the two seasons, as shown in Appendix A.

### 4.3. Field Experiments and Environmental Conditions

These diverse genotypes were evaluated following a transplanting method under 4 different N-levels (Urea form) to generate the different environments of no N (0N), Low N (LN), medium N (MN), and High N (HN) (i.e., 0, 48, 96, and 165 Kg N ha^−1^, respectively). The experimental design of each environment was employed according to the randomized complete block design (RCBD) with three replicates. Each plot was five one-meter-long rows, with 20 cm × 20 cm spacing. N fertilizer was applied in two splits: 2/3 as a basal application and 1/3 a month after transplanting according to the N fertilizer amount for each N environment. Other cultural practices (such as field preparation, fertilizers, and weed management) were applied according to the standard to maintain the consistency of all factors except N-levels across the studied environments. At the harvest stage, the plots were harvested, and GY (g plant^−1^) was estimated for each experimental plot after adjusting the moisture content.

### 4.4. Statistical Analysis

All statistical analyses were conducted in the statistical software R (R Core Team, [55]) version 4.1.1. The “metan” package [37] was employed to conduct the analysis of variance, AMMI analysis of variance [43], genotype plus genotype by environment (GGE) biplot analysis [56], stability statistical analysis, and weighted average of absolute scores [50]. The “Nbclust” package was used to conduct hierarchical cluster analysis [57,58]. The AMMI analysis was conducted based on the following mathematical formula:$$y_{ij}^{N}=\;\mu +{\;g}_{i}+{\;e}_{j}+{\;\Sigma \lambda }_{k}Y_{ik}\alpha_{jk}+{\;\varepsilon }_{ij}$$
where *y_ij_* is the yield of the *i*th genotype in the *j*th environment, *N* is the number of PCI in the AMMI model, μ is the overall mean of the genotypes, and g*_i_* and e*_j_* are the genotype and environment diversions from the overall mean. λ_k_ is the eigenvalue of the PCA axis *k*, Y*_ik_*and α*_jk_* are the GE-PCs scores for axis *k,* and Σ*ij* is the remaining value. Meanwhile, the GGE model was considered by the following formula:$$y_{ij}^{N}=\;\mu +e_{j}+{\;\Sigma \lambda }_{k}Y_{ik}\alpha_{jk}+{\;\varepsilon }_{ij}$$

## 5. Conclusions

In the current investigation, 55 rice genotypes were evaluated under 4 different N- levels to identify the genotypes with high yields under low N input and with stable performance under different N-levels. The results showed that the highest-ranked genotypes were IR22, SK2046, SK2058, MTU1010, YABANI LULU, and IR66 based on the GY mean. These genotypes were grouped in the same SCL according to the stability analysis ranking matrix. Based on AMMI analysis and biplots, MTU1010 and YABANI LULU showed stability, while the which-won-where biplot showed that IR22 was superior under unfavorable N-levels and MTU1010 was stable across the different environments. The study recommends incorporating these genotypes into the breeding for low-nitrogen fertilizer input programs for sustainable agriculture.

## Figures and Tables

**Figure 1 plants-11-02775-f001:**
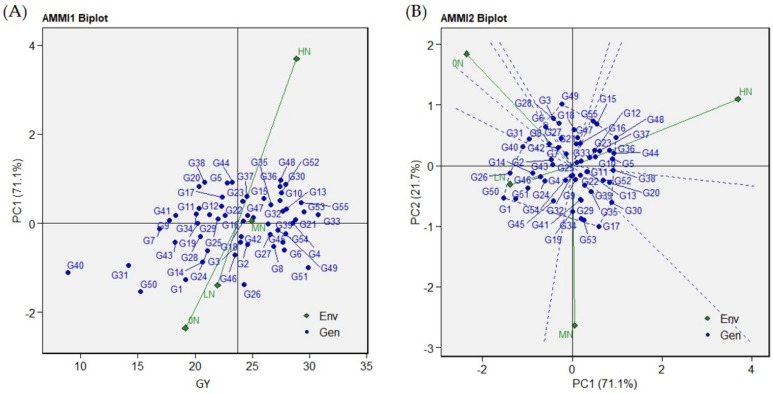
The AMMI1 and AMMI2 (**A**,**B**), respectively) biplots indicating the GEI for the 55 rice genotypes across 4 N environments. The genotypes’ and environments’ legends are presented in Table 2.

**Figure 2 plants-11-02775-f002:**
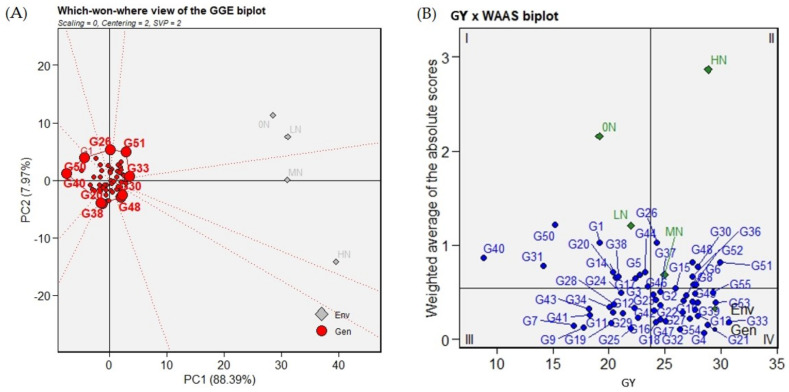
Which-won-where pattern based on the GGE biplot polygon to identify the best rice genotypes under the four different N environments (**A**). The GY × WAAS statistic biplot for selecting the high-yielding and stable rice genotypes (**B**).

**Figure 3 plants-11-02775-f003:**
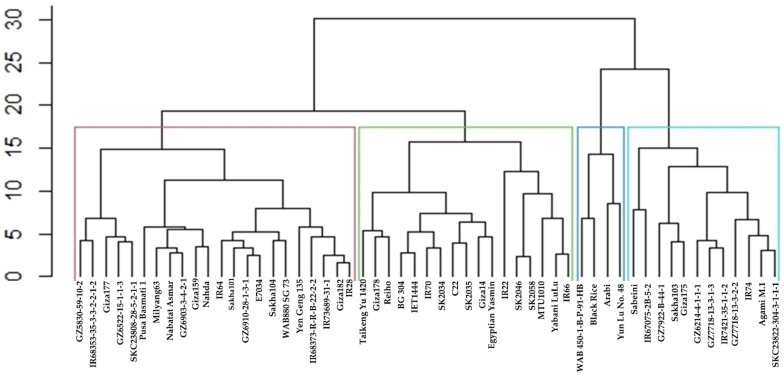
Hierarchical classification dendrogram of the 55 tested rice genotypes based on their ranks for GY and stability statistics via Ward’s method. SCL 1 (light blue), SCL 2 (Blue), SCL3 (green), and SCL 4 (red).

**Table 1 plants-11-02775-t001:** AMMI analysis based on GY means for genotypic performance and N environment EPC1 value and the 5 top ranking genotypes for each N environment.

E	Mean	EPC1	Index	Class	1	2	3	4	5
0N	19.13842	−2.35395	−4.575242	Unfavorable	IR22	GIZA14	C22	MTU1010	SK2058
LN	21.94194	−1.39131	−1.771727	Unfavorable	IR22	MTU1010	Yabani Lulu	SK2035	IR66
MN	24.93612	0.054521	1.222455	favorable	SK2046	IR22	MTU1010	Egyptian Yasmin	Giza178
HN	28.83818	3.690741	5.124515	favorable	SK2058	MTU1010	Reiho	WAB 880 SG 73	Giza178

**Table 2 plants-11-02775-t002:** Name, type, parentage, and origin of the plant materials.

No.	Genotype	Type	Parentage	Origin
1	Sabieny	J	Selection from Introductions	EGYPT
2	Nabatat Asmar	J	Selection from Agami M1	EGYPT
3	Giza 159	J	Giza14/Agami M.1	EGYPT
4	Yabani LuLu	J	Selection from Introductions	EGYPT
5	GZ 5830-59-10-2	J	GZ4120/Suweon349	EGYPT
6	Giza 14	J	Yabani Pearl/Iraki16	EGYPT
7	GZ 7718-13-3-1-3	J	Sakha101/HR4856-1-1-2	EGYPT
8	Nahda	J	Selection from Introductions	EGYPT
9	GZ 6214-4-1-1-1	J	GZ4122-23-4-2/IRI396	EGYPT
10	Sakha 101	J	Giza176/Milyang79	EGYPT
11	IR 68373-R-R-B-22-2-2	T.J.	JINMIBYEO/YR14987-91	IRRI
12	Giza 182	I	Giza181/IR39422//Giza181	EGYPT
13	GZ 7922-B-44-1	J	Giza177/IDSA	EGYPT
14	Pusa Basmati 1	I	India selection	INDIA
15	Sakha 104	J	GZ4096/GZ4100	EGYPT
16	IR 28	I	IR8333-6-2-1///IR1561-149-1//IR24*4/O. NIVARA	IRRI
17	GZ 6522-15-1-1-3	J	GZ5581/GZ4316	EGYPT
18	IR 64	I	IR5657-33-2-1/IR2061-4665-1-5-5	IRRI
19	GZ 7718-13-3-2-2	J	Sakha101/HR4856-1-1-2	EGYPT
20	Giza 177	J	Giza171/Yamji No.1//PI NO.4	EGYPT
21	IR 66	I	IR13240-108-2-2-3/IR9129-209-2-2-2-1	IRRI
22	GZ 6910-28-1-3-1	J	Sakha101/GZ24316_(MUT)_	EGYPT
23	IR 70	I	IR19660-73-4/IR54//IR9828-36-3	IRRI
24	Agami M.1	J	Selection from cultivated varieties	EGYPT
25	Sakha 103	J	GZ4120/Suweon349	EGYPT
26	Arabi	I/J	Java3/Yabani Montkhab 3	EGYPT
27	Milyang 63	I/J	TONGIL/IR946-33-2-2-2//YR675-131-2	KOREA
28	Yen Geng 135	J	Chinese selection	CHINA
29	IR 73689-31-1	T.J.	SR18977-TB-4/JINMIBYEO	IRRI
30	Giza 178	I/J	Giza175/Milyang49	EGYPT
31	WAB 450-1-B-P-91-HB	I	---	Africa Rice
32	BG 304	I	---	SRILANKA
33	MTU 1010	I	---	INDIA
34	IR 68353-35-3-3-2-2-1-2	T.J.	CHEOLWEON49/KYWHA9	IRRI
35	Giza 175	I/J	(IR28/IR1541)/(Giza180/Giza14)	EGYPT
36	WAB 880 SG 73	I	---	Africa Rice
37	E 7034	J	EWAN NO.5/857	CHINA
38	SKC 23808-28-5-2-1-1	I/J	98-Y-116/Sakha102	IRRI
39	IET 1444	I	TN1/CO.29	INDIA
40	Black Rice	J	Jingo9601	China
41	IR 7421-35-1-1-2	T.J.	IR2035-290-2-1-1/MASINO	IRRI
42	GZ 6903-3-4-2-1	J	Sakha101/Suweon313	EGYPT
43	SKC 23822-304-3-1-1-1	I/J	M202/Giza177	IRRI
44	Taikeng Yu 1420	J	C253///J692130/BL6//TAINUNG67/IR4547-2-1-2	TAIWAN
45	Egyptian Yasmine	I	IR262-43-8-11/KDML105	EGYPT
46	IR 67075-2B-5-2	I	IR10198-66-2//GZ2175/CSR1	IRRI
47	IR 74	I	IR19661-131-1-2/IR15795-199-3-3	IRRI
48	Reiho	J	HOYOKU/AYANISHKI	JAPAN
49	C 22	I	TJRERMAS/BPI76//PALAWAN/AZUCENA	IRRI
50	Yun Lu No. 48	J	LUYIN NO.7/YUNANJINGDAO-38	CHINA
51	IR 22	I	PETA/DEE GEO WOO GEN//TADUKAN	IRRI
52	SK2034	I	IR69625A/Giza178R	EGYPT
53	SK2046	I	IR69625A/Giza181R	EGYPT
54	SK2035	I	IR70368A/Giza178R	EGYPT
55	SK2058	I	IR69625A/Giza182R	EGYPT

J, Japonica; I, Indica; T.J., Tropical japonica; I/J, Indica japonica; IRRI, International Rice Research Institute.

## Data Availability

Data are available in the Appendix A.

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
