# Peer review of "Detection of Superior Rice Genotypes and Yield Stability under Different Nitrogen Levels Using AMMI Model and Stability Statistics"

_plants, 2022, doi:10.3390/plants11202775_

Round 1
Reviewer 1 Report
The authors identified superior genotypes from a large pool of genotypes that offer stable yields under variable N levels or function better under low N conditions.
Major remarks:
1. Avoid identifying genotypes based on the numbers you assign. Make use of the real identification.
2. Mention the year(s) and season(s) during which the study was conducted.
3. Explain why you chose to provide yield per plant rather than yield per unit area. Explain the setup of the experiment. The plot size appears to be too small for full randomization.
4. It's unclear whether the study was timely repeated.
5. Offer some quantitative yield measurements as well.
6. Give more information about the relevant genotypes.
Author Response
Resposne attached

Reviewer 2 Report
This manuscript is for genetic analysis of salt tolerance for Bangladesh costal landrace. The pehotype evaluation and genetic maker analysis were well conducted. I think this manuscript has useful information to rice breeders for looking for alt tolerant germplasm.
This manuscript can be published but need to be revised.
I would advise the authors to re-read the manuscript and rectify the typographical errors in the manuscript.
line 46 ~ line 48 : need ref.
There is not Figure 1.
I suggest the table 1 move to the supplemention.
Define IPCA, GNI and PC at first mention of each
Give full definitions for all abbreviations (ENV, REP, GEN, IPCV, DF, Sq, LN, MN, HN, E and EPC1, alos in supplementary table) in the table legend or as a footnote to the table, even has mentioned on main paragraph.
In Figure 2 but first figure in article, author have to indicate seperatly as like Figure 3 (actually 2nd figure) a and b. Therefor, decript like as a: AMMI1 Biplots, b: AMMI2 Biplot.
I wonder weather the ecotype( japonica, tropical japonica, indica and indica*japonica ) was fitted the genotype properly or not. If author have genotyping data or reference paper, should present in this MS. And i want to know how the genome structure of these genetic resources was.
Reviewer 3 Report
The authors need to see comments or suggestions (yellow highlighted & comment boxes) made within the attached manuscript. They need to include all suggestions.

Author Response
Have been responsed
Round 2
Reviewer 1 Report
The manuscript has been improved to my satisfaction.